# Design and Implementation of Cloud Docker Application Architecture Based on Machine Learning in Container Management for Smart Manufacturing

**Byoung Soo Kim [1], Sang Hyeop Lee [2], Ye Rim Lee [3], Yong Hyun Park [3] and Jongpil Jeong [1],***

[1] Department of Smart Factory Convergence, Sungkyunkwan University, 2066, Seobu-ro, Jangan-gu, Suwon 16419, Korea; jmk9996@gmail.com
[2] Department of System Management Engineering, Sungkyunkwan University, 2066, Seobu-ro, Jangan-gu, Suwon 16419, Korea; sanghyeop96@g.skku.edu
[3] Department of Mechanical Engineering, Sungkyunkwan University, 2066, Seobu-ro, Jangan-gu, Suwon 16419, Korea; dldpfla1024@g.skku.edu (Y.R.L.); east1ifeP@gmail.com (Y.H.P.)
* Correspondence: jpjeong@skku.edu; Tel.: +82-31-299-4267

**Abstract:** Manufacturers are expanding their business-process innovation and customized manufacturing to reduce their information technology costs and increase their operational efficiency. Large companies are building enterprise-wide hybrid cloud platforms to further accelerate their digital transformation. Many companies are also introducing container virtualization technology to maximize their cloud transition and cloud benefits. However, small- and mid-sized manufacturers are struggling with their digital transformation owing to technological barriers. Herein, for small- and medium-sized manufacturing enterprises transitioning onto the cloud, we introduce a Docker Container application architecture, a customized container-based defect inspection machine-learning model for the AWS cloud environment developed for use in small manufacturing plants. By linking with open-source software, the development was improved and a datadog-based container monitoring system, built to enable real-time anomaly detection, was implemented.

**Keywords:** cloud docker; docker container; machine learning; monitoring; smart manufacturing; container management

## 1. Introduction

The digital transformation of the traditional manufacturing industry has been delayed compared to other industries owing to time and cost. Meanwhile, with the outbreak of COVID-19, manufacturers have been directly affected by issues in the overall manufacturing value chain, including production, supply, and distribution. To overcome such crises and enhance competitiveness, we are preparing for a new leap into digital transformation. Many manufacturing companies conduct maintenance through server management and by using monitoring systems in an on-premises-based IT infrastructure environment. However, the complexity of IT services and surges in network traffic are creating inefficiencies that continually increase costs. Manufacturing companies are considering moving to a public cloud as an alternative for efficient server management and adapting to the rapidly changing manufacturing IT infrastructure. The introduction of container-based virtualization technology is essential for maximizing the benefits of cloud usage, including scalability, cost-efficiency, and global coverage. With the recent and continuous development of container-based virtualization environments, server administrators (hardware/application/service) can run multiple operating systems on the same hardware simultaneously, making it possible to also package and distribute various applications. Supporting continuous development and improving the efficiency and reliability of runtime environments, container-based virtualization is gaining popularity in many different areas [1].

In addition, as the connection among containers continues to increase, sophisticated management and analysis must be conducted to prevent additional problems from occurring. Thus, a monitoring tool that supports the reliable management and security of containers has been proposed. However, it is difficult to select a lightweight, open-source-based container application and a monitoring tool with excellent portability [2].

Docker is the most representative lightweight virtualization technology for container platforms. It can be used to package web applications as Docker images and run them on any cloud host that has a Docker execution environment. Web application deployment has become increasingly convenient and flexible. There is no need to rely on a single provider, and one can easily migrate web applications between different cloud providers, avoid a lock-in by the providers, and take advantage of competitive pricing markets [3]. Docker provides the ability to package and run applications in loosely isolated environments, called containers. Isolation and security allow many containers to operate concurrently on a given host. Containers are lightweight and contain everything required to run an application; therefore, there is no need to depend on current installations on the host. Containers can easily be shared while working, and everyone with sharing access will have the same containers that behave in the same way [4]. As another reason for choosing Docker, it provides an open-source-based Docker Node Visualizer to visualize user-friendly container nodes, and it has a community edition that allows managing Docker resources (e.g., container, images, volume, and network) through Portainer. An administrative web UI is also provided. In addition, the deployment, management, scaling, and networking of containerized applications are automated through Docker Swarm Orchestration, and computational resources, including hardware resources (e.g., CPU, memory, storage, and I/O), can be efficiently managed and easily accessed through the cloud. They can also be relocated or moved [5].

The contributions of this paper are as follows:

1. By learning a container-based machine-learning application and building a defect inspection system, we aim to lower the barriers to entry into a digital transformation for small- and medium-sized manufacturers.
2. We hope to help improve the quality of application building/distribution services (time/CPU/memory) for the use versus non-use of containers.
3. We aim to contribute to container life-cycle management by predicting real-time anomalies and failures through container-monitoring management tools and visualizations.

The remainder of this paper is organized as follows. Section 2 describes the related operations. Section 3 describes the proposed container-based architecture and all of its components. Section 4 describes the experimental progress, evaluation indicators, and results. Section 5 presents some concluding remarks and areas of future research.

## 2. Related Work

### 2.1. Docker Container

Docker [6] is currently the most widely used container platform [7]. Because Docker containers do not virtualize hardware, they are much lighter and faster. As shown in Figure 1, Docker containers can run on small devices to large servers, and on average, containers run 26× faster than virtual machines (VMs) [7,8]. According to the survey [9], approximately 25% of organizations have adopted Docker technology. According to a survey conducted by DataDog in early April 2018, the percentage of hosts running Docker containers has continued to increase. According to a survey conducted in April 2018, 21% of all hosts used Docker containers. Since 2015, the share of customers running Docker has grown at a rate of about 3 to 5 points per year [10].

The survey also noted that usage rates have increased as the underlying host infrastructure has grown in size [9]. Among the organizations with 1000 or more hosts, 47% have adopted Docker technology, whereas only 19% have fewer than 100 hosts [10]. The advent of Docker containers is providing a transition into an alternative to traditional VMs. Many studies have compared containers and VMs using Docker as the container technology. Containers use a Linux kernel mechanism for resource allocation. When creating a container, any user can allow the allocation of resources, such as the network configuration, CPU, and memory. Although allocated resources can be dynamically adjusted, containers cannot use more resources than specified [9].

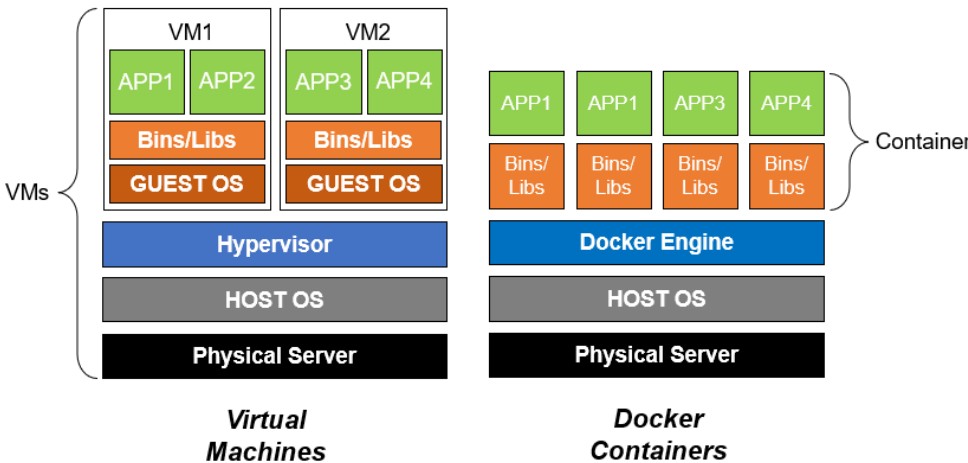

**Figure 1.** Virtual machines vs. containers.

*2.2. Docker Container Management Tool*

Container technology is primarily used for the background execution of programs without a graphical user interface (GUI). However, many applications require GUI support to enable interaction with the user [11]. For the Visualizer [12] tool for Docker Swarm, there is a project available on Github [13], the Docker Swarm Visualizer. This project is visible to the user of the host container. However, it does not provide insight into overlay networks [11]. Docker applies a poor platform monitoring approach. By default, the only type of monitoring solution that Docker provides is the stat command. This is appropriate only if extremely basic container information, not advanced monitoring, is needed [14].

Figure 2 shows Portainer, which is used to manage containerized Nginx web servers and can be used with Kubernetes, Docker, Docker Swarm, and other services, providing significant benefits for building better self-hosted data centers. By providing a GUI instead of the Docker graphical user interface, i.e., a command line interface (CLI), one can quickly deploy, manage, and observe the behavior of the web server containers and provide appropriate and immediate security whenever needed [15]. Portainer builds additional features for developers and infrastructure teams, such as application management, cluster management, registry/image management, identity and access management, storage management, network management, monitoring, and alerts [16].

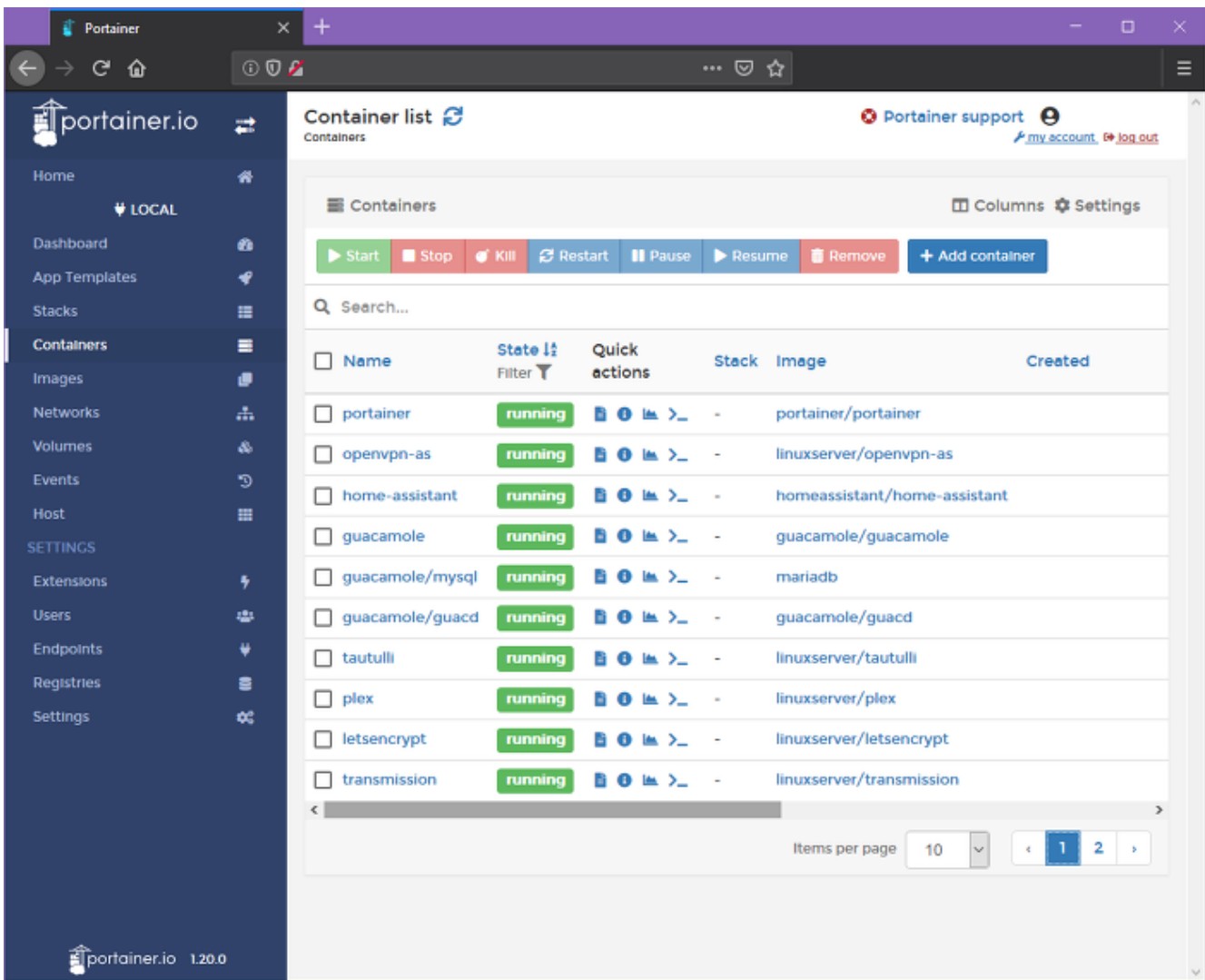

**Figure 2.** Portainer GUI for Docker container management.

### 2.3. Machine Learning

A support vector machine (SVM) is a new machine-learning method proposed by Vapnik [17,18]. This method follows the principles of structural risk minimization and limited sample assumptions and can overcome some of the weaknesses of conventional machine learning, such as those of neural networks. For example, SVMs can overcome convergence problems and other high-level disaster problems. Therefore, SVMs are good at learning and are more likely to be generalized. SVMs are one of the youngest branches of statistical learning theory and can be expressed in a form similar to that of a neural network. Statistical learning was evaluated as the best theory for predictive research and statistical estimation with a small sample size. We view learning as a general process of estimating functions based on empirical data. An SVM follows the principles of structural risk minimization and provides a new cognitive perspective for machine learning. Traditional neural networks first modify the trust risk and then minimize the heuristic risk. An SVM adopts the opposite method to correct the empirical risk, minimize the risk of confidence, and then map the input space to a high-dimensional dot space; thus, the risk is independent of the number of samples entered, thereby avoiding the input dimension and hence, the "curse of dimensionality." By solving the linear constrained quadratic programming problem, an SVM can obtain a global optimal solution without the local minima problem, and the fast algorithm guarantees the speed of convergence. Through the above process, the structural parameters are automatically determined from the sample; thus, an SVM can overcome the



shortcomings of a conventional neural network and be used as a general-purpose learning machine. The early warning of a financial crisis is a process of pattern recognition. SVMs are well-suited for pattern recognition, making them applicable to warnings regarding a financial crisis [19].

A linear discriminant analysis (LDA) is a supervised ML technique used for extricating the significant highlights from a dataset. To limit the computational cost, an LDA is used to avoid overfitting the information. This is accomplished by anticipating a feature space onto a slightly lower-dimensional space with optimal class detachability. With an LDA, more accentuation is given to those axes that are responsible for maximizing the segment among various classes [20].

K-nearest neighbors (k-NN) is a robust and flexible classifier that belongs to the supervised ML family of algorithms. Because it has no explicit assumptions regarding the distribution of the dataset, a k-NN is a nonparametric algorithm. The algorithm stores every accessible case and classifies new cases based on a similarity measure. A case is classified by the dominant part of the vote of its neighbors, with the case being appointed to the more generally regular class among its k-nearest neighbors estimated based on a distance function [21,22].

Recent research efforts to improve the reliability and accuracy of image classification have led to the introduction of the Support Vector Classification (SVC) scheme. SVC is a new generation of supervised learning methods based on the principle of statistical learning theory, which is designed to decrease uncertainty in the model structure and the fitness of data [23].

The bootstrap forest is an RF approach that uses an ensemble of classification trees by averaging many decision trees, each of which is fit to a bootstrap sample of the training data. Each split in each tree considers a random subset of the predictors. In other words, given a training set S of n examples, a new training set S0 is constructed by drawing m examples uniformly (with replacement) [24].

The boosted tree (gradient boosting) approach maintains a set of weights over the original training set S and adjusts these weights after each classifier is learned by the base learning algorithm. The weights of examples that are misclassified are increased by the base learning algorithm, and the weight of examples that are correctly classified are decreased [25].

The naïve Bayes (NB) classifier is a machine-learning algorithm that greatly simplifies learning by assuming that features are independent given class, that is, $P(X|C) = \prod \sum_{i=1}^{n} P(X_i|C)$, where $X = (X_1, \cdots, X_n)$ is the vector of features, and C is the class. Naïve Bayes classifiers assign the most likely class to a given subject depending on its feature vector. Although it includes unrealistic assumptions, NB is considered remarkably successful in practice compared to other more sophisticated algorithms. Its applications include medical diagnosis [26] and food quality classification [27].

As shown in Table 1, we will explain the advantages and disadvantages of algorithms used in machine learning training.

**Table 1.** Advantages and disadvantages of machine-learning algorithms used.

| Algorithm | Advantages | Disadvantages |
|---|---|---|
| SVM/SVR | High performance, high accuracy, good handling of high dimensional data [17,28,29] | Lack of transparency in high dimensional data, extensive memory requirements [17,28,29] |
| linear discriminant analysis | Low computational cost Easy to implement Discriminate different groups Visual representation makes clear understanding [27,30] | Requires normal distribution Linear decision boundaries Limited to two classes [27,30] |
| K-Nearest Neighbors | Intuitive and simple, easy to implement for multiclass problems [31,32] | Computationally expensive in large datasets, performance depends on dimensionality [31,32] |
| Artificial Neural Networks | Good at handling large datasets, detect all possible interactions between prediction variables, implicit detection of complex non-linear relationships between dependent and independent variables [33,34] | High hardware dependencies (GPU), Unexplained behavior of the network, the duration of the network is unknown [33,34] |

### 3. Cloud Docker Application Architecture Based on Machine Learning

*3.1. System Architecture*

We developed a lightweight virtualization technology based on Docker containers that can be run in any cloud environment. We propose a cloud docker architecture, shown in Figure 3. This makes the deployment and testing of machine-learning applications convenient and flexible, and by linking them with a highly portable management tool, monitoring and container management become convenient.

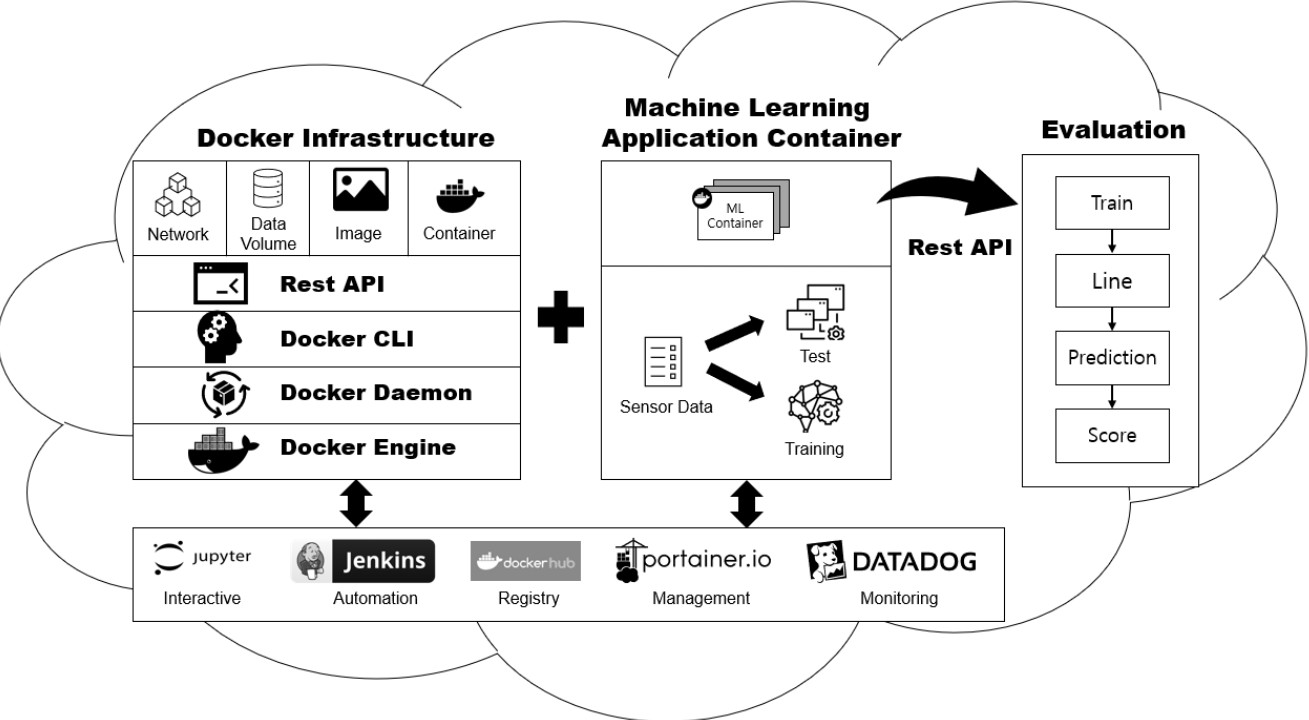

**Figure 3.** Docker Container-based machine-learning pass/fail inspection system on a cloud server and a docker-based monitoring system architecture.

Our study separates a large-scale system in a cloud server environment into application container units, making it easy to build, deploy, and test as an independent service unit and builds an extremely effective environment for standardizing the application operation, increasing the code and resource usage. In addition, a prototype of a Docker container-based machine-learning fault determination system is built on a cloud server, and an architecture is proposed that can detect anomalies in real-time and prevent failures through container-monitoring tools and visualizations. Docker Container, which is the most widely used among small- and medium-sized container platforms, can run the same container anywhere, including desktops, virtual machines, physical servers, data centers, and clouds and provides specific supplies such as AWS, Azure, GCP, and Oracle Cloud, which are major public clouds. It is provided in a platform-as-a-service form, which is not dependent on the vendor.

*3.2. Docker Container*

As shown in Figure 4, applications create, control, and manage containers through Docker Daemon (network, data volume, image, and container) of the Docker Engine. Although Docker Daemon operates on the host machine, the operator does not directly touch the Daemon, and the Docker Client interacts with the Daemon through the Docker CLI. A Docker File is a text file configured to build a Docker Image. Using commands, image files are added and copied, commands are executed, and ports are exposed. The Docker

file is configured from the base image declaration to start the process command. Several command types are provided [35].

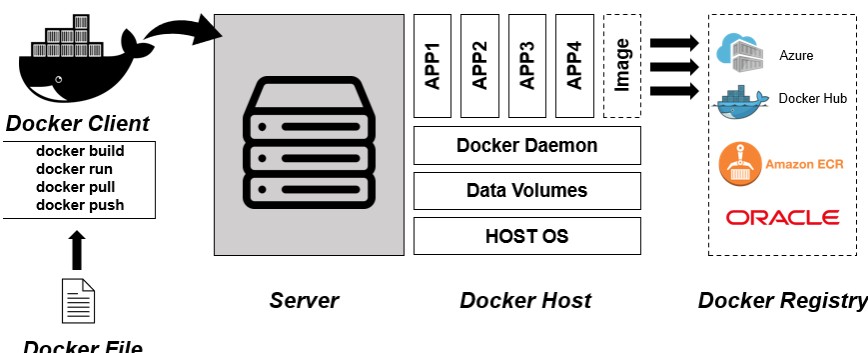

**Figure 4.** Docker Container key features.

After completing the Docker file setting, the Docker Client can create an image through the Docker build command. An application packaged as an image is Docker's Union File System, which is a set of "layers" and consists of file and directory. The data volumes are the data part of the container and are initialized when the container is created. The volume can maintain and share container data, and even if the container is destroyed, updated, or rebuilt, it remains as is. If it needs to be modified, it must be done so directly [35,36].

The Docker Registry is an open-source-based Apache repository that stores and distributes images. It is highly scalable, provides image storage location control and a distributed pipeline, and can be integrated with local development workflows. Users can set up their own registry or a hosted one such as Docker Hub [37], Amazon ECR [38], Oracle Container Registry [39], or Azure Container Registry [40].

A Docker Container can be considered an image execution instance that contains all elements, such as the OS, development source code, runtime, system lib, and system binary, required to run an application packaged as an image. Open-source-based solutions are installed and linked to build a cloud development environment (DevOps implementation) [41].

*3.3. Container Infrastructure*

A monolithic software design does not allow for scalability of the architecture and cannot manage different levels of load at runtime. For this reason, all components that constitute the architecture have been integrated as microservices inside the container infrastructure to enable scalability, high availability, and both vertical and horizontal migration. Container virtualization technology has received significant attention in the past few years owing to such features as a fast container-building process, high density of services per container, and high isolation between instances [42]. Unlike traditional hypervisors, a lightweight virtualization technology implements process virtualization through the containers in an operating system. This allows the deployment of high-density containers by reducing the hardware overhead and the virtualization of virtual appliances on traditional hypervisors [5].

The next Docker component is a machine-learning container-based defect inspection system and open-source-based DevOps tools [41] for container monitoring implementation.

- Portainer: this is a Docker-paper used to manage the Docker clusters and Docker resources (e.g., containers, images, volumes, and networks). Portainer is an administrative web UI with a community edition that makes it easy to manage Docker clusters without writing multiple lines of script code [5].
- Jenkins: this is a popular Java-based server tool for automation with the help of plugins. Jenkins is considered a powerful application that helps automate software development processes through continuous integration and the delivery of papers,

regardless of the platform being worked on [43]. It is automated to build and deploy machine-learning-based defect inspection applications and push them to the Docker Hub for container image management.

- Docker Hub: this is the largest group of container images available in the world. Images on Docker Hub are organized into repositories, which can be divided into official and community repositories. For each image in a Docker Hub repository, in addition to the image itself, meta-information is also available to the users, such as the repository description and history, in a Dockerfile [37].
- Jupyter Notebook: this is mainly used for service development for interactive computing across open-source software, open standards, and multiple programming languages. Jupyter Notebook supports the Julia, Python, and R programming languages. Jupyter Notebooks can potentially revolutionise the documentation and sharing of research software towards an unprecedented level of transparency for relatively low effort [44].
- DataDog: this is a monitoring service that collects metrics, such as the CPU utilization, memory, and I/O, for all containers. An agent-based system that only transmits data to the DataDog cloud makes monitoring operations completely dependent on this cloud [45].
- Docker Swarm Visualizer: this is an open-source paper that provides a user-friendly web UI for visualizing nodes belonging to a Docker cluster and containers deployed on such nodes [5].

## 4. Implementation and Results

### 4.1. System Configuration

The implementation environment was configured for experimentation, as shown in Figure 5. The AWS cloud environment used for testing was implemented using both hardware and software, as shown in Table 2. The main purpose of this experiment was to measure and compare the CPU overhead, execution time of the machine learning, and memory usage. The analysis results with and without the Docker Container are shown.

**Table 2.** Cloud server computing environments.

| Item | Resource |
| --- | --- |
| Cloud | Amazon Web Service |
| Region | ap-northeast-2 |
| Service | EC2 |
| OS | Amazon Linux |
| Kernel | Linux |
| Instants type | T2.medium |
| Key Pairs | RSA |
| CPU | 2 |
| Storage | SSD(gp2) 25G |
| MEM | 4G |

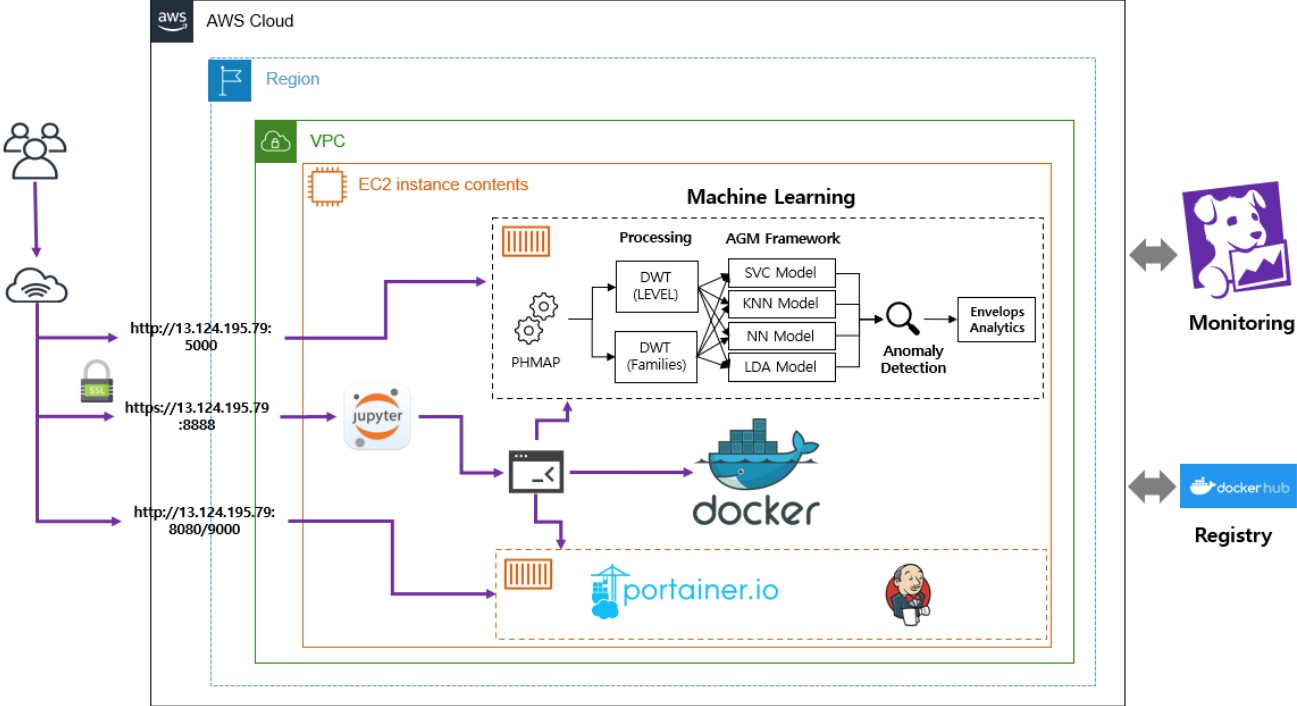

**Figure 5.** Docker Container-based machine-learning pass/fail inspection system on AWS cloud server environment and a docker-based monitoring system, and image registry interconnection.

Jupyter Notebook: as shown in Figure 6, the convenience of the CLI is enhanced by registering the Jupyter Notebook with Jupyter.service for running automatically when Amazon EC2 boots.

**Figure 6.** Jupyter Notebook Interactive.

As shown in Figure 7, Jenkins is a tool that automates a build distribution, and the user can check the execution time of the batch jobs. Jenkins is automated to run the Docker Build/Run—a machine-learning defect determination application shown in Figure 8—and push it to the Docker Hub server, which is an image registry [37].

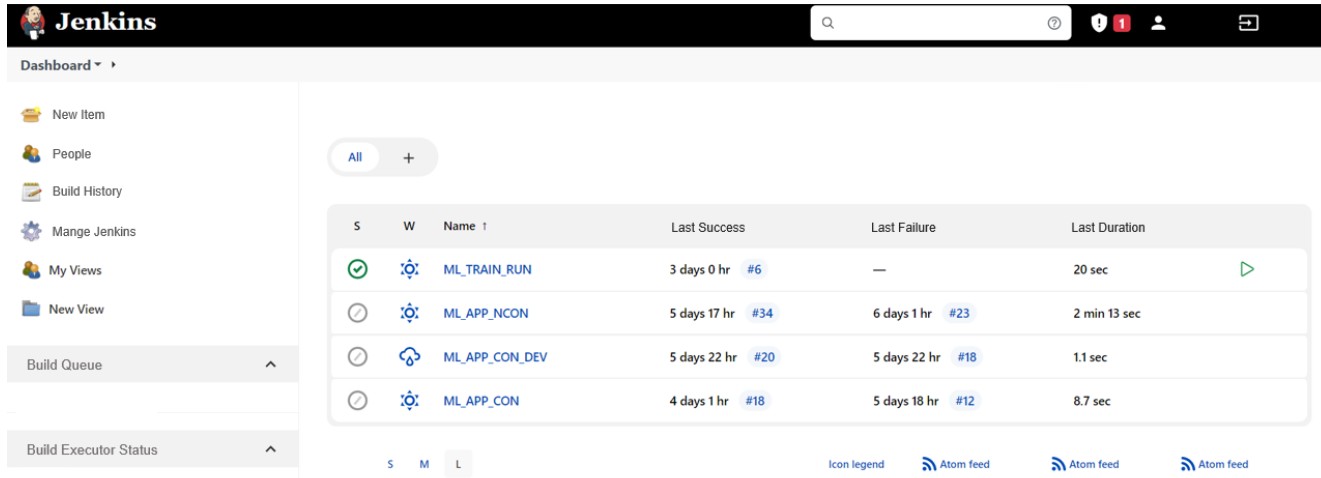

**Figure 7.** Automation job list and the time duration of number of builds.

```
echo "*********start1*******"
date '+%F  %r'

DOCKER_BUILDKIT=1 docker build -t smartfactory_capstone -f Dockerfile .
docker rm -f smartfactory_capstone || true

docker tag smartfactory_capstone jmk9996/smartfactory_capstone
docker push jmk9996/smartfactory_capstone

docker run --name smartfactory_capstone -d -p 5000:5000 smartfactory_capstone python3 api.py

date '+%F  %r'
echo "*********end*******"
```

**Figure 8.** Docker build/run and Docker Hub push.

As shown in Figure 9, the Docker Hub—a Docker image registry—is a repository that hosts, indexes, and manages images.

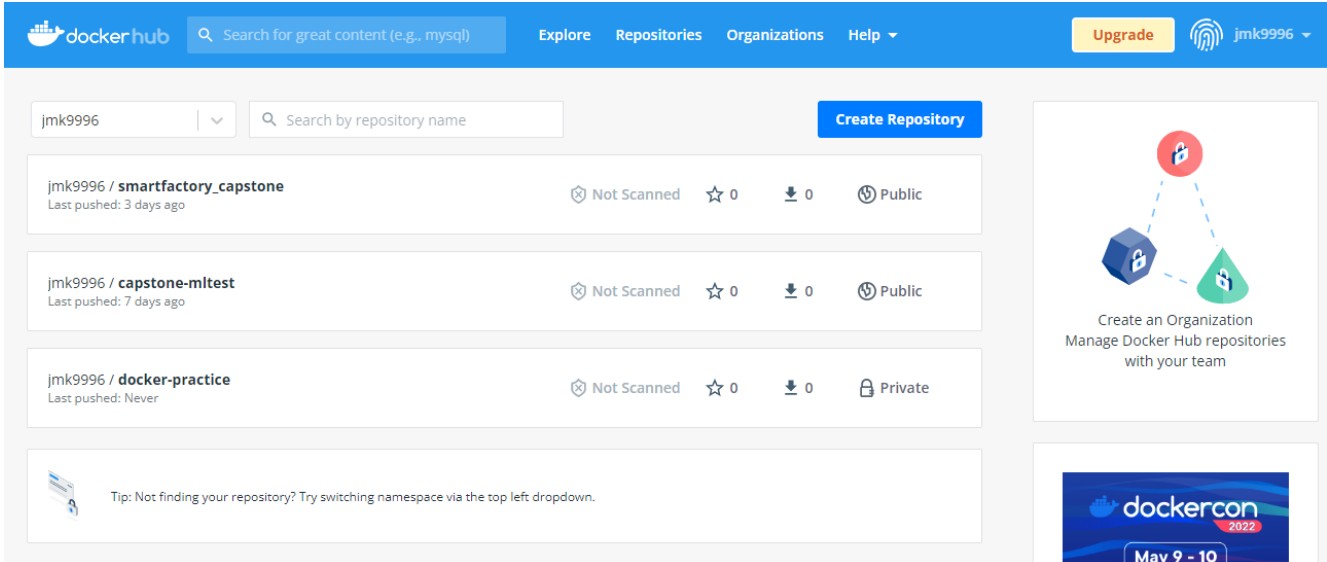

**Figure 9.** Docker Hub registry for Docker image management.

### 4.2. Dataset and Machine-Learning Model

The target data used in the machine-learning modeling shown in Figure 10 are the time-series data for the state of a two-channel bearing. This problem must be classified into a total of four modes (one normal and three defective), and 30 feature values were

extracted through feature extraction (data source: PHMAP 2021 Asia Data Challenge). Four machine-learning models were trained on the above data. The machine learning used lightweight models such as a support vector classifier (SVC) [23], linear discriminant analysis (LDA) [20], neural network (NN) [30], and K-nearest neighbor (KNN) [21,22].

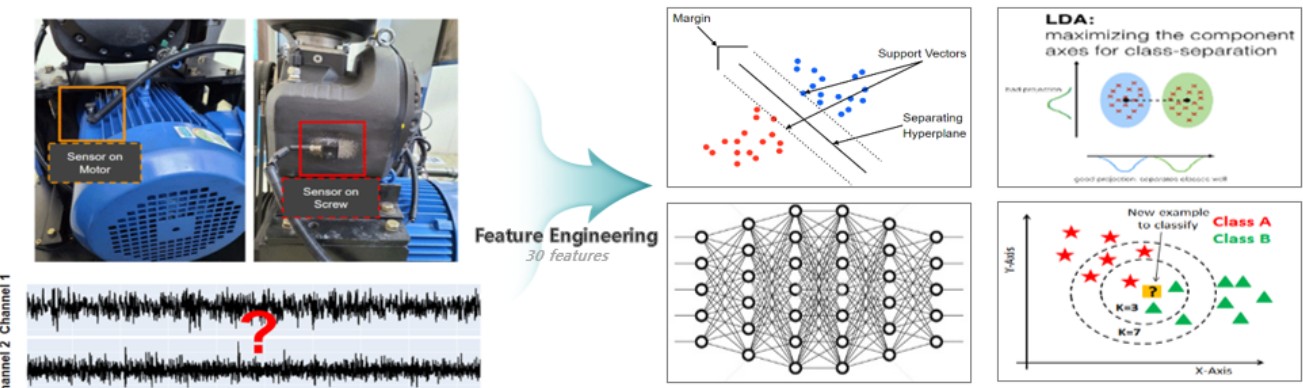

**Figure 10.** Machine-learning modeling.

### 4.3. Docker Container-Based Defect Inspection System

As shown in Figure 11, the implementation goals and scenarios are divided into three phases. First, the machine-learning model is packaged as a Docker image and is then built and run as a Docker container. Monitoring is conducted using a Datadog.

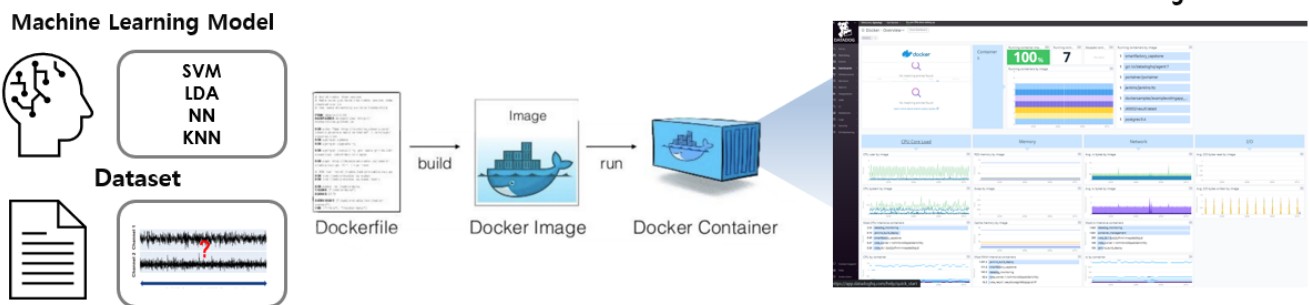

**Figure 11.** Machine-learning model is packaged as a Docker image, and run as a Docker container.

The main function of the Docker Container, i.e., Rest-Api, is implemented. First, the model training is conducted at the same time Docker is executed, and when the user inputs three commands, i.e., line, prediction, and Evaluation (score), to the web server, the corresponding result is output. When outputting the data, Prediction outputs the prediction results of the four models described above, whereas Evaluation (score) outputs the overall accuracy of the test set for the currently trained model.

- csv_to_json.py: the DataSet field consists of Line, defect, and feature1 30, and then converts the .csv file into a .json format.
- Api.py: learning is applied using four types of machine-learning models. In this study, four ML algorithms were applied to classify the test datasets. The SVC, LDA, NN, and KNN models were selected to solve the quaternary classification problem rather than binary classification. They were then combined.

Figure 12 shows the documentation of the machine-learning-based Rest-API.

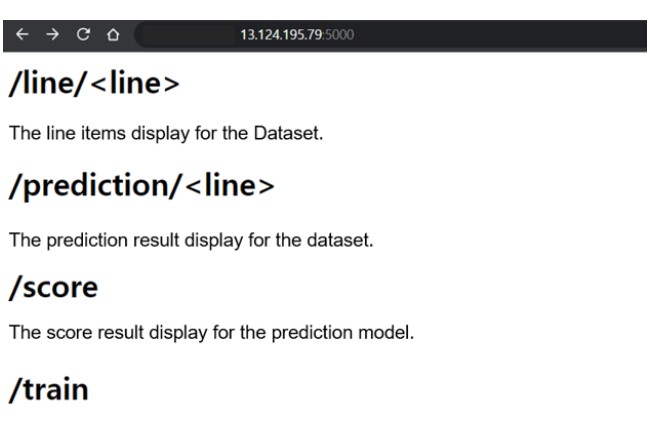

**Figure 12.** Documentation for REST-API.

As shown in Figure 13, we built a monitoring system by linking DataDog to the monitoring part, which is a weakness of the Docker Container-based platform. DataDog has been proven to be a comprehensive cloud monitoring service for the Docker Container life cycle and server resource management used in a cloud environment.

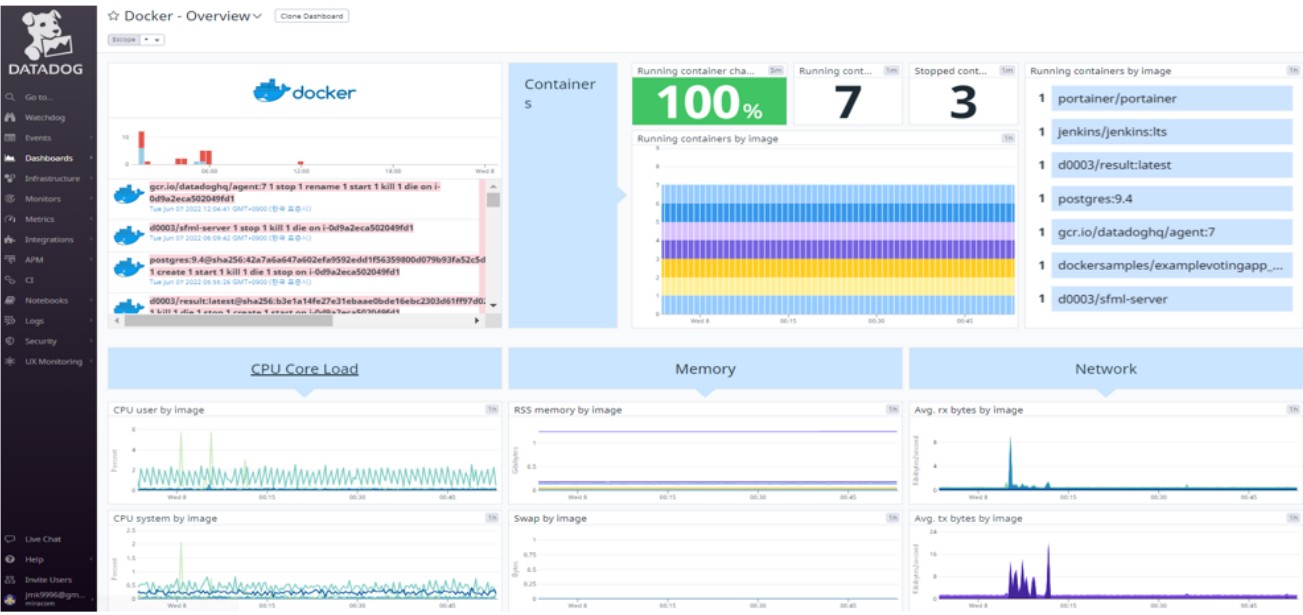

**Figure 13.** Cloud Docker based monitoring system.

### 4.4. Results

The system performance was tested using a machine-learning application. The system performance was checked under various loads with and without the use of Docker.Machine-learning applications, and containers were measured by expanding the number of containers from a minimum of one to a maximum of four. These measurements included the application container memory usage, application execution time, memory usage during execution, and network I/O block in the scenarios shown in Figures 14–16. The ratio of heads per container with and without the use of Docker is displayed and verified.

As shown in Figure 14, when Docker Container is not used, the CPU overhead increases from a minimum of 150% to a maximum of 191%, and when Docker Container is used, the optimal performance increases from a minimum of 1% to a maximum of 7%. This shows that Docker Container can handle a load while reducing the CPU usage, allowing the device to operate optimally without loss.

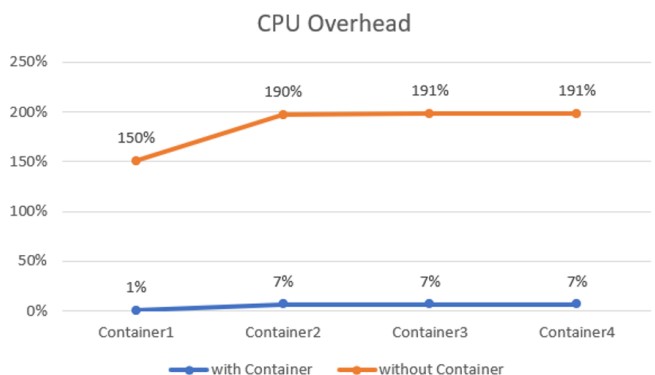

**Figure 14.** The graph representations of CPU overhead and CPU utilization with and without Docker.

As shown in Figure 15, when Docker Container is not used, the execution time is increased from 33 s in the case of one container to 124 s in the case of four, whereas when Docker Container is used, the execution time is 30 s for one container, which does not differ much from the case when Docker Container was not used. However, when expanding the number of containers to four, outstanding results were measured with a marked difference from when Docker Container was not used. It was proven that running applications in a lightweight Docker Container-based virtualization environment is effective in terms of both time and cost.

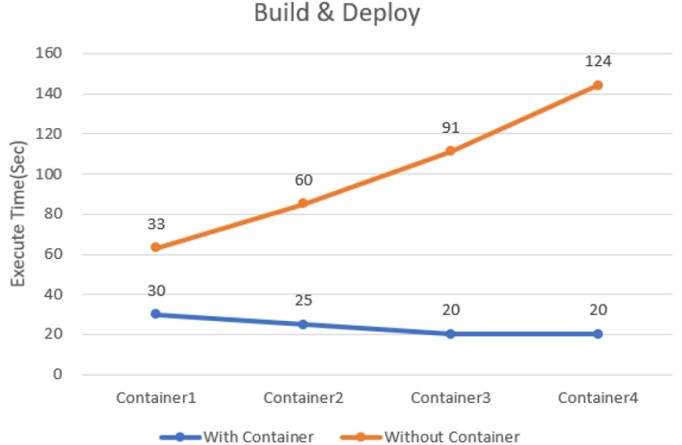

**Figure 15.** The Graph representations of execution times for builds and deployments with and without Docker.

Figure 16 confirms that there is no significant difference in terms of memory usage. However, it was verified that the memory usage rate in the cumulative execution part—similar to the execution time—shows a gradually stable usage rate when Docker Container is used. The container has a much lighter operation than the existing virtualization technology, shares the OS kernel, and uses significantly less memory. This is thought to be due to the IT operation team operating a large number of systems, which has many advantages because it uses resources in a unit with much less memory usage.

We built a prototype of a Docker Container-based machine-learning pass/fail inspection system on a cloud server and a data-docker-based monitoring system for Docker containers. All of our papers are open-source and can be easily implemented; therefore, it is expected that the barriers to entry for manufacturing companies undergoing a digital transformation will be lowered. In addition, it will be possible to provide customized applications according to the specific circumstances of each company through microservices rather than the existing monolithic structure for the necessary functions. Monitoring results also confirm the high-level resource management performance of Docker containers.

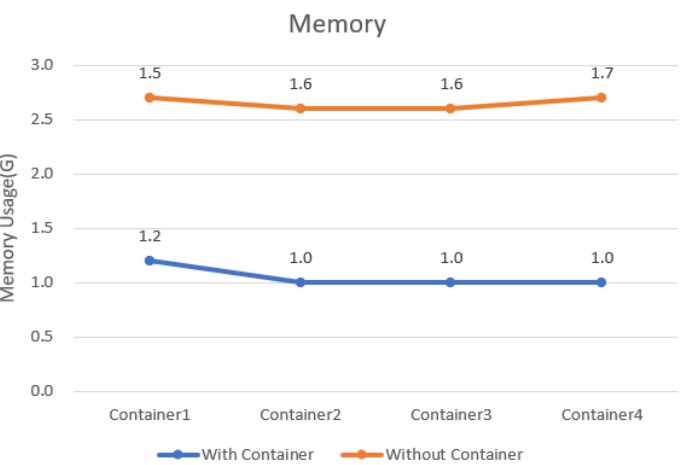

**Figure 16.** The Graph representations usage of memory with and without Docker.

## 5. Conclusions

We propose a Docker Container detect inspection system optimized for training and running machine-learning models in a cloud environment. By building automated environments with open-source software, users can easily customize the machine learning, build and deploy Docker containers, and quickly obtain the results. The monitoring part—which is a weakness of container technology—can also use DataDog to monitor the health of the container and cloud server resources in real-time. Through our research, it is expected that the cloud container machine-learning-based defect inspection system will accelerate the digital transformation of small and medium-sized manufacturers and contribute to the improvement of service quality by building container applications. In addition, it will be of great help in predicting equipment abnormalities and managing container life cycle through open-source-based container monitoring tools and visualizations. A real-time data receiver can be implemented in a real factory if future architectural developments are implemented.

In addition, it seems that the architecture can be further improved by linking the Grafana dashboard to strengthen the monitoring, improve the container service quality through a Docker orchestration, and increase the development and operational efficiency through GitHub integration.

**Author Contributions:** Conceptualization, B.S.K. and J.J.; methodology, Y.R.L.; software, S.H.L. and Y.H.P.; validation, Y.H.P. and S.H.L.; formal analysis, Y.R.L. and Y.H.P.; validation, S.H.L. and J.J.; investigation, B.S.K.; resources, J.J.; data curation, Y.R.L.; writing—original draft preparation, B.S.K.; writing—review and editing, J.J.; visualization, S.H.L. and Y.R.L.; supervision, J.J.; paper administration, J.J.; funding acquisition, J.J. All authors have read and agreed to the published version of the manuscript.

**Funding:** This research was supported by the MSIT (Ministry of Science and ICT), Korea, under the ITRC (Information Technology Research Center) support program (IITP-2022-2018-0-01417) supervised by the IITP(Institute for Information & Communications Technology Planning & Evaluation). Also, this work was supported by the National Research Foundation of Korea (NRF) grant funded by the Korea government (MSIT) (No. 2021R1F1A1060054).

**Institutional Review Board Statement:** Not applicable.

**Informed Consent Statement:** Not applicable.

**Data Availability Statement:** Not applicable.

**Acknowledgments:** This research was supported by the MSIT(Ministry of Science and ICT), Korea, under the ICT Creative Consilience Program(IITP-2022-2020-0-01821) and the ITRC(Information Technology Research Center) support program(IITP-2022-2018-0-01417) supervised by the IITP(Institute for Information & communications Technology Planning & Evaluation).

**Conflicts of Interest:** The authors declare no conflict of interest.

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
