# Peer review of "Design and Implementation of Cloud Docker Application Architecture Based on Machine Learning in Container Management for Smart Manufacturing"

_applsci, doi:10.3390/app12136737_

Round 1

Reviewer 1 Report

20220615 1st review report

The article is interesting and generally, it deserves to be published with some revisions that are suggested below:

1)     Would you please move the contributions of this article from Introduce to the Conclusion?

2)     You need to introduce the "Neural networks for machine learning", and what are the advantages and disadvantages while practicing since you mentioned it?

3)     You may try to introduce Bootstrap Forest, Boosted Tree, and Naive Bayesmain for main  Machine Learning, in part 2 (Related Work).

4)     Would you please refer to your study results, and describe in the Abstract, for example, to build a prototype of a Docker container-based machine learning pass/fail judgment system on a cloud server, and also built a data dock-based monitoring system for Docker containers.

Reviewer 2 Report

The paper Present a good topic. However, there is some points which I would like the authors invest more time to improve their paper.

-         - Abstract should be organized more logically, some ideas can be moved to introduction section

-       -  Line 40 - 41 should be deleted (from you can to execution environment).

-        - Introduction need to be improved and reformulated to ease the reading, also add more references and try to avoid general statement   

-       -  Some parts need to add references, for example authors provide the background of SVM model without any reference, this is not allowed, at least they need to cite the author who developed the algorithm for the first time.

Good Luck.

Reviewer 3 Report

My comments are as follows

The article is quite interesting, but requires some minor changes:

- drawings number 10 (especially the right part of the drawing), 11, 12 and 13 should be made again, because they are completely illegible,

- in the case of bibliographies, some literature items, e.g. 12 or 15, do not have a year of publication, and in principle there is no access date to the publication

- in general, the list of literature is very haotic, it should be adapted to the requirements of the editorial office

Round 2

Reviewer 2 Report

The authors addressed all my previous comments from the first version of the manuscript, Congratulations to the authors for this nice job.